# Paper-Based Humidity Sensor for Respiratory Monitoring

**DOI:** 10.3390/ma15186447

**Published:** 2022-09-16

**Authors:** Xiaoxiao Ma, Shaoxing Zhang, Peikai Zou, Ruya Li, Yubo Fan

**Affiliations:** 1Key Laboratory for Biomechanics and Mechanobiology of Ministry of Education, Beijing Advanced Innovation Center for Biomedical Engineering, School of Biological Science and Medical Engineering, Beihang University, Beijing 100083, China; 2Department of Otolaryngology-Head and Neck Surgery, Peking University Third Hospital, Beijing 100191, China; 3School of Engineering Medicine, Beihang University, Beijing 100083, China

**Keywords:** paper-based humidity sensor, respiratory monitoring, paper humidity sensing model, oral breathing, nasal breathing, sleep-related respiratory disorders

## Abstract

Flexible respiratory monitoring devices have become available for outside-hospital application scenarios attributable to their improved system wearability. However, the complex fabrication process of such flexible devices results in high prices, limiting their applications in real-life scenarios. This study proposes a flexible, low-cost, and easy-processing paper-based humidity sensor for sleep respiratory monitoring. A paper humidity sensing model was established and sensors under different design parameters were processed and tested, achieving high sensitivity of 5.45 kΩ/%RH and good repeatability with a matching rate of over 85.7%. Furthermore, the sensor patch with a dual-channel 3D structure was designed to distinguish between oral and nasal breathing from origin signals proved in the simulated breathing signal monitoring test. The sensor patch was applied in the sleep respiratory monitoring of a healthy volunteer and an obstruct sleep apnea patient, demonstrating its ability to distinguish between different respiratory patterns as well as various breathing modes.

## 1. Introduction

Respiration has been recognized as one of the most basic physiological signals to assess the physiological condition of the human body [1]. Respiratory status is essential for disease diagnosis and evaluation of sleep-related respiratory disorders, such as obstructive sleep apnea (OSA) and oral breathing, which can cause deformities in the mouth and maxillofacial region [2,3,4,5,6]. Thus, respiratory monitoring devices are of great practical significance for these applications. As the gold standard for clinical sleep respiratory monitoring, polysomnography (PSG) can comprehensively and accurately assess patients’ breathing conditions by monitoring multiple physiological signals, including electrocardiogram, electroencephalogram, respiratory airflow, and eye movements [7,8]. Meanwhile, portable devices for daily breathing monitoring appeared in different situations outside the hospital, such as masks for CO_2_ concentration monitoring, in-ear microphones detecting sounds in the ear canal, and nose-mounted devices with PPG measurement units and accelerometers to detect sleep breathing events [9,10,11,12,13]. Although these portable devices mentioned have enabled respiratory monitoring on an outside-hospital daily basis, their wearability and comfort are still concerns in the application scenarios [14]. Thus, devices based on flexible sensors appeared to improve the wearability of the respiratory monitors and increase the comfortability between the device–skin interface, such as chest straps and sleep mattresses based on flexible piezoelectric sensors and the skin-like hybrid integrated circuits [15,16,17,18,19]. However, most flexible respiratory monitoring devices still rely on complex microfabrication processes, which add to the cost of each device.

As a flexible substrate material, paper has some characteristics of low cost, water absorptivity, and degradability, which brings new possibilities to the flexible sensing field. Therefore, it has been adopted recently to replace the traditional substrate material of flexible electronic devices [20,21,22]. For instance, paper-based microfluidic chips can be used for clinical or chemical analysis by designing the capillary force of the paper-based microfluidic channel [23]. The production of paper-based sensors usually includes methods such as screen printing and roll-to-roll printing, which lower the production cost and bring broad prospects for the paper to be incorporated into industrial mass-producible flexible electronic devices [24,25,26,27].

In this article, we introduce a paper-based humidity sensor (PHS) based on the hygroscopic characteristics of paper for flexible respiratory monitoring. PHS has the specialties of flexibility, low price, easy processing, as well as the ability to distinguish between nasal and oral breathing signals by a designed dual-channel 3D structure. By establishing the paper humidity sensing model of paper material, the principle that the PHS can sense the humidity variations is elaborated, as Figure 1a shows the microscopic diagram of paper fiber and the water layer attached to it, while the water layer thickness increases (or decreases) as the ambient humidity rises (or drops), indicating the microcosmic mechanism of the PHS. In terms of the sensing performance, tests have been carried out to prove the high sensitivity up to 5.45 kΩ/%RH and 411 Ω/%RH in low (58%RH to 75%RH) and high (75%RH to 100%RH) humidity levels, respectively. Repeatability in cyclic measurement has also been compared to standard relative humidity with a matching rate of over 85.7%. Besides humidity sensing properties, experiments have been conducted to certify the ability of PHS in respiratory monitoring applications. By monitoring simulated breathing signals, the capability of PHS to respond to breathing modes with different respiration rates and amplitudes as well as distinguish the breathing signals of the mouth and nose from the original signal through a dual-channel structural design has been proved. Finally, sleep respiratory monitoring has been realized on a healthy volunteer and an OSA patient using the flexible device throughout a two-hour sleeping test. Both oral and nasal breathing signals have been monitored and used to classify sleep breathing into four distinct respiratory patterns, including oral breathing, nasal breathing, oral–nasal co-breathing, and no-breathing, during which the breathing event occurs for the OSA patient. This device brings a novel, affordable sensing strategy for clinicians and patients with sleep-related respiratory disorders, which provides more accessible respiratory information during disease monitoring, diagnosis, and treatment processes.

## 2. Materials and Methods

### 2.1. Sensor Fabrication

The printing paper used to build the PHS was purchased from Double A, Thailand. Printing paper of three different grams per square meter (gsm) was chosen, including 70 gsm, 75 gsm, and 80 gsm. The graphite ink (MCJ-518, Qianhai JiShengya Technology Company, Shenzhen, China) was screen-printed to the printing paper through the designed interdigital electrode pattern. The PHS patch with the designed 3D structure was processed, as the detailed fabrication process is shown in Figure 1b. For the purpose of distinguishing between oral and nasal respiratory signals, two graphite screen electrodes were printed separately on both sides of the printing paper. The graphite ink electrodes were baked dry on a heating plate at 80 °C for 30 min. After die-cutting, wire connecting, pasting, and folding, a dual-channel PHS patch with a 3D spatial structure was made. Figure 1c shows the picture of the PHS patch.

### 2.2. Sensor Calibration Setup

In order to characterize the humidity sensing behavior of the paper-based sensor, we built an experimental calibration platform. The commercial temperature and moisture meter Testo 635-2, with a measuring range of 0 to 100%RH and accuracy of ±2%RH at 25 °C, was purchased from Testo Technology Company, Lenzkirch, Germany. It was used in the experiment to measure relative humidity and environment temperature within the experimental settings.

To carry out the humidity calibrating experiment, the PHSs, as well as the standard humidity meter, were placed in a closed chamber. The humidifier was used to create unidirectional humidity changes within the cavity. Step humidity increase and decrease were set, and the corresponding resistance of PHS was measured when reaching equilibrium at different relative humidity conditions.

The sensor response time was tested by placing the sensor into the humidity chambers with set humidity levels to evaluate the time needed for reaching equilibrium. PHS was transferred from low environmental relative humidity of 28%RH to high humidity environments (70%RH, 80%RH, and 91%RH). After the resistance readout stabilized in the high humidity level, PHS was moved back to the low humidity environment with the recovery time recorded.

In the repeatability test, the standard humidity meter probe and PHS were placed adjacent in front of the periodically operating humidifier, which generated 30 times of humidity variation with a 4 min period in the environment by ejecting an equal amount of moisture into the sensor. The resistance of the sensor and environmental humidity were measured simultaneously.

In the simulated breathing signal monitoring experiment, various respiratory patterns and breathing modes were produced artificially while signals were monitored by oral and nasal sensors simultaneously. In the sleep respiratory monitoring test, one healthy volunteer and one OSA patient were recruited. The designed 3D PHS patches as well as the flexible PCBs were used to detect signals during an over-two-hour sleep.

### 2.3. Signal Acquisition System

An acquisition system (shown in Figure 2) was designed for respiratory monitoring experiments, including sensors, flexible printed circuit board for signal acquisition and transmission, and PC-based signal processing. The resistance changes of the two sensors were converted into voltage by two identical resistance measurement circuits. Then, the circuit output signals were sampled by the ADC module with a frequency of 18 Hz, and the MCU transmitted signals to PC through the BLE module for further processing. The whole flexible PCB was powered by a 3.7 V lithium battery, making it more convenient for portable use.

A preprocessing filter was performed on the acquired signal so as to get rid of the noise introduced by the circuit. The raw signal was filtered by a median filter with a window width of 15, and then a smooth filter with a window width of 9 before further signal processing.

## 3. Results

### 3.1. Paper Humidity Sensing Model

To explain the phenomenon that the resistance of paper-based sensors changes with relative humidity, a paper humidity sensing model is established. Paper, as a short fibers composite, consists mainly of cellulose and hemicellulose, which are both hydrophilic materials. When exposed to high relative humidity, water molecules tend to condense and form a water layer on the surfaces of the fibers. The salts left on the fibers after paper processing, such as sulfates, dissolve in water and come into ions, which conduct through directional movement in the solvent. When the ambient humidity decreases, the condensed water on the fibers evaporates and quickly diffuses into the atmosphere. We establish a paper humidity sensing model of the printing paper based on water condensation theory as well as interactions between particles and surfaces theory [28,29]. The paper fiber is considered to remain unchanged under different relative humidity levels, while the change in resistance is caused by the thickness variation of the condensate water layer on the fiber. Thus, according to the laws of resistance combining the interdigital electrode structure, the relationship between the paper-based sensor resistance *R* and the thickness of the aqueous layer on the fiber *t* is shown as Equation (1)
(1)$$R=\frac{\rho dr^{2}}{LhA_{f}\left({\left(r+t\right)}^{2}-r^{2}\right)}$$
where *d* and *L* are the interfinger gap width and interfinger gap length of the interdigital electrode, *h* is the thickness of the paper, *r* is the average radius of the paper fiber, *A_f_* is the fiber area fraction of the paper composite, and *ρ* is the resistivity of the water layer. For the sensor constructed with interdigital electrodes, the width and gap of the fingers are both 0.5 mm, forming a 7 mm × 7.8 mm sensing area. For the interdigital electrodes, the number of fingers changes the resistance of the sensor by varying the overall interfinger gap length *L*. According to Equation (1), both interfinger gap length *L* and paper thickness *h* change the resistance range of sensor responding to relative humidity, which further leads to variations in sensitivity. After weighing the sensitivity and device resistance range, appropriate *L* and *h* for measurement were determined. Based on Majumdar’s theory of water condensation on hydrophilic surfaces as well as Israelachvili’s study of the interactions between particles and surfaces, the relationship between relative humidity and the thickness of the water layer is shown in Equation (2)
(2)$$\psi =\exp\left\{-\alpha \left[\frac{Ue^{-t/\lambda }}{\lambda }(1+\frac{t}{\lambda }+\frac{\lambda^{2}}{t^{2}})+64\beta e^{-\kappa t}-\delta \frac{1}{t^{3}}-\eta e^{-t/\lambda }\right]\right\}$$
where *ψ* is the relative humidity, *λ* is the length-scale parameter in hydration energy, and *U* is the free-energy parameter in hydration energy [28,29]. Specific meanings of the constant coefficients *α*, *β*, *δ*, *η*, together with detailed formula derivations, can be found in the Appendix A. By combining Equations (1) and (2), the paper humidity sensing model is established, which illustrates the relationship between the resistance of the paper-based sensor and the relative humidity of the environment, as shown in Figure 3a.

According to our paper humidity sensing model, the thickness of the water film on the surface of fiber *t* increases with the increase of relative humidity *ψ*, as indicated by Equation (2). Based on Equation (1), an increase in *t* leads to a decrease in the resistance *R* of the paper-based sensor. In terms of the changing trend, the change rate of resistance to the relative humidity in the low humidity zone is much higher than that in the high humidity zone.

### 3.2. Humidity Sensing Performance on the Paper-Based Humidity Sensor

Sensitivity was characterized under the experimental setting detailed in the Methods section and the result of the relationship between relative humidity and resistance of PHS is shown in Figure 3b. Three types of printing paper with different gsm were tested in 58%RH, 71%RH, 77%RH, 82%RH, 92%RH, and 99%RH environments. A rapid resistance decrease can be observed in all PHSs with different parameters as the relative humidity level increases. Compared with the PHS with lower gsm, the resistance of the sensor using 80 gsm paper changes more severely with a sensitivity of 5.45 kΩ/%RH, as humidity changes in the low humidity range of 58%RH to 75%RH. In a high humidity range of 75%RH to 100%RH, the sensor using 80 gsm paper also exhibits good sensitivity of 411 Ω/%RH. Hysteresis characteristic of the PHS in 80 gsm was also analyzed by measuring PHS resistance within a controlled humidity level variation between 58% and 99%RH, as shown in Figure 3c.

For respiration monitoring purposes, the high sensitivity to humidity enables the PHS to detect different respiration patterns with refined signals. The PHS with 80 gsm printing paper shows sufficient sensitivity and small hysteresis error within the targeted sensing range of ambient relative humidity to 100%RH to cover the reported exhale breath relative humidity of over ambient to 91%, for the application in respiratory sensing [30]. Moreover, compared to 70 gsm and 75 gsm printing paper, 80 gsm paper is thicker, making it more favorable for maintaining the designed spatial 3D structure in practical use. Considering the sensing range, sensitivity, hysteresis factors, and the strength for maintaining the dual-channel 3D structure, PHSs made of 80 gsm paper are chosen for further tests and respiration monitoring applications.

The response and recovery characteristics of the 80 gsm sensors are as shown in Figure 3d. The sensors respond immediately to humidity change, and a higher output voltage changing rate is observed when the relative humidity changes more drastically. However, at the same time, the response time also turns out to be more extended, which increases from 382 s for a 70%RH environment to 991 s for a 91%RH case. The recovery time increases from 22 s for 70%RH to 48 s for 91%RH, which is generally much shorter than the response time. Generally, the response time is tested to be long, which is at least 382 s for humidity increasing and 22 s for humidity decreasing under experimental conditions. However, in contrast to the saturation status of the sensor at different humidity levels in the response time test, the sensor is actually non-saturated when responding to the periodically varying respiratory humidity signal. Within 3–5 s of a breathing cycle, a transient response with sufficient amplitude occurs in the sensor to reflect the user’s breathing waveform.

The designed cyclic test to verify sensor repeatability was carried out by simulating the periodic moisture flow during breathing. As shown in Figure 3e, the paper-based humidity sensor can detect humidity changes repeatedly. The amplitude of the humidity waveform acquired by the paper-based sensor matches well with that calibrated by the standard hygrometer. To quantitatively evaluate the degree of matching of sensor signals with periodic ambient humidity changes, the relative matching rate *M* is proposed, which is defined as *M* = (*PPV*/*PPV*_0_)/(*PPRH/PPRH*_0_), where *PPRH*_0_ and *PPV*_0_ are the peak-to-peak values of standard humidity and voltage output from the sensor measurement circuit in the first cycle, and *PPRH* and *PPV* are that of other cycles. The matching rate *M* remains over 85.7% during the repeatability experiment, which shows good repeatability of PHS.

### 3.3. Respiratory Monitoring of Simulated Breathing Signal

To distinguish the oral and nasal breathing signals, a paper-based sensor patch was attached to the skin between the nasal septum and upper lip to collect oral and nasal breathing signals separately, forming a dual-channel PHS working mode, as shown in Figure 4a. In the simulated breathing signal monitoring experiment, the volunteer was asked to breathe consciously and regularly, following the set respiratory patterns and breathing modes, which makes the simulated breathing signal closer to real breathing conditions.

Firstly, three different breathing modes, including normal breathing, deep breathing, and rapid breathing, were simulated and tested. According to Figure 4b, the breath waveform amplitude reflects the breathing depth, and the waveform frequency corresponds to the breathing rate. Signals under different breathing modes within the normal respiratory rate of 12 to 20 breaths per minute can be well detected and differentiated [31].

The ability to distinguish between oral and nasal breathing was then characterized, as shown in Figure 4c. Signals were recorded in four respiratory patterns, including nasal breathing, oral breathing, oral–nasal co-breathing, and no-breathing, corresponding to Region I, II, III, and IV, respectively. The oral and nasal airflows in four respiratory patterns are shown in Figure 4d. In the nasal pattern, a significant breathing signal is detected through the nasal sensor, while the breathing signal of the oral sensor is barely measured. In the oral-breathing period, the situation is contrary. As both oral and nasal breathings are involved, breathing signals can be detected by two sensors, showing a smaller amplitude than that of oral or nasal breathing alone. In the no-breathing pattern, the no-breathing signal is measured from either the oral sensor or the nasal sensor, demonstrating the ability of the PHS patch to detect apnea events. In the experiment, the crosstalk between the oral and nasal sensors is calculated to be less than 12.2%. As certified by simulated breathing signal experiments, the flexible device based on PHS can detect and differentiate different breathing modes of various frequencies and depths.

### 3.4. Sleep Respiratory Monitoring

There are many different human breathing patterns during sleep, which vary significantly in breathing depth and respiratory rate. For people with sleep-related respiratory disorders, abnormal breathing, including apnea and excessively shallow breathing, may be present during their sleep. In order to verify the sensor’s response and monitoring capabilities to complex sleep breathing signals, sleep breath monitoring tests were conducted on a healthy volunteer and an OSA patient.

A paper-based sensor patch was stuck on the skin, and other modules, including a flexible PCB and battery, were placed in the sleep mask to increase the wearability of the device and reduce body motion artifact during sleep. The sleep respiratory monitoring experiments were conducted over two hours with each participant. The result of the breathing signal is shown in Figure 5 and Figure 6. Figure 5a shows the breathing signal of the healthy volunteer during the two-hour monitoring, including nasal breathing (I), oral breathing (II), and oral–nasal co-breathing (III) modes. The amplitude of the respiration waveform remains overall stable during two hours of monitoring, without frequent switching of respiratory patterns. Moreover, the sleep respiratory rate is maintained within the normal range of 12–20 breaths per minute. Detailed signals for different sleeping stages from ① to ④ are also illustrated in Figure 5b–e. From stage ① and stage ④, when the volunteer breathes with the mouth or nose alone, the corresponding sensors are able to detect oral and nasal signals separately, with no visible signal from the other sensor detected. According to the signals in stage ② and stage ④, breathing signals of different amplitudes at various breathing depths can be distinguished. Stage ③ illustrates the ability of the device to monitor the oral–nasal co-breathing signal, differentiating oral breathing signal from nasal directly. During 41′30″ to 63′, a drift signal is observed, which is caused by changes in the volunteer’s sleeping body position. When switching to a particular position, the ambient humidity around the volunteer’s mouth and nose increases, causing a baseline drift that quickly disappears when the position returns to normal. The periodic breathing waveforms remain detectable regardless of the body position change.

Figure 6a shows the respiration monitoring result of an OSA patient during the over-two-hours test. Suffering from OSA, the patient’s airway collapses during sleep, resulting in hypopnea and even apnea. As a result, oral breathing may be adopted as a complement to oxygen that cannot be obtained by nasal breathing alone. This behavior is reflected in the breath waveform, as shown in Figure 6b–d. Compared with the healthy volunteer, the OSA patient significantly tends to use oral–nasal co-breathing. In the process of oral–nasal co-breathing stage ⑤ and stage ⑥, oral and nasal respiratory signals do not maintain a stable amplitude and frequency. Instead, dramatic fluctuations in both the oral and nasal signals have been observed. In addition, a sleep apnea event is identified during the two hours of breath monitoring. The breathing event begins at around 58′45″, as shown in Figure 6d, when the breath waveform disappears in both the oral and nasal signals, indicating the occurrence of sleep apnea. After the apnea event lasted two minutes, oral and nasal breathing resumes almost simultaneously at 60′45″.

The sleep respiratory monitoring experiments have proved that the wearable device using the designed dual-channel paper-based humidity sensor has the ability to distinguish between the mouth and nose breathing from the original signal and further differentiate various respiration patterns. Besides, the device can not only monitor the regular sleep breathing signals of a healthy volunteer but also detect the cluttered breathing signals of a patient with sleep-related respiratory disorders. Respiration of the healthy volunteer is mainly composed of regular oral breathing, nasal breathing, and mouth–nose coordinated breathing, while oral–nasal co-breathing accounts for a significantly higher proportion for the OSA patient. The device can be a helpful tool for breathing event detection, which can provide doctors with more information on the sleep-related respiratory disorders diagnosis and treatment process.

## 4. Conclusions

In summary, a flexible, highly sensitive PHS with good stability has been developed for sleep respiratory monitoring. With low cost and simple processing, the paper-based sensor can be regarded as a disposable part of the flexible device, making it portable and hygienic for daily usage. The 80 gsm PHS shows high sensitivity (5.45 kΩ/%RH in low humidity level and 411 Ω/%RH in high humidity level) as well as good cyclic performance in the device characterization process. The designed dual-channel 3D structure enables the sensor to distinguish oral and nasal breathing signals directly from the original signal. Moreover, experiments on long-term respiratory monitoring during sleep have been carried out, in which we observed the characteristics of sleep respiratory signals in a healthy volunteer and an OSA patient, proving that the use of PHS in oral and nasal breathing monitoring can provide more information about how gas flows through different airways. The device may provide healthcare workers with more breathing information about patients with sleep-related respiratory disorders to assist in the diagnosis and treatment process.

## Figures and Tables

**Figure 1 materials-15-06447-f001:**
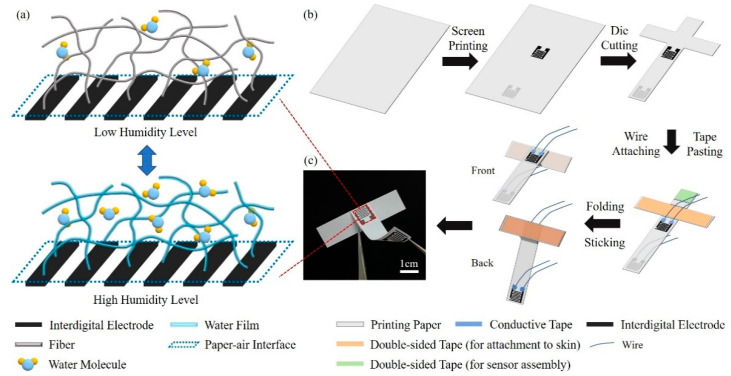
(**a**) Humidity sensing principle of the PHS. (**b**) Processing of designed 3D structure sensor patch. (**c**) Photo of the flexible PHS.

**Figure 2 materials-15-06447-f002:**
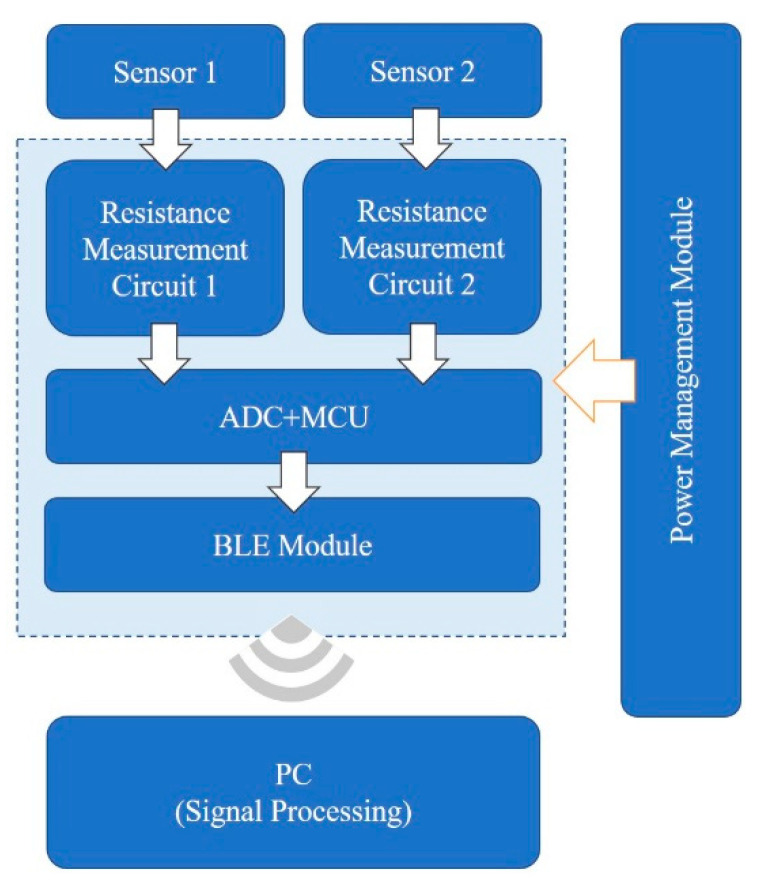
System flow diagram.

**Figure 3 materials-15-06447-f003:**
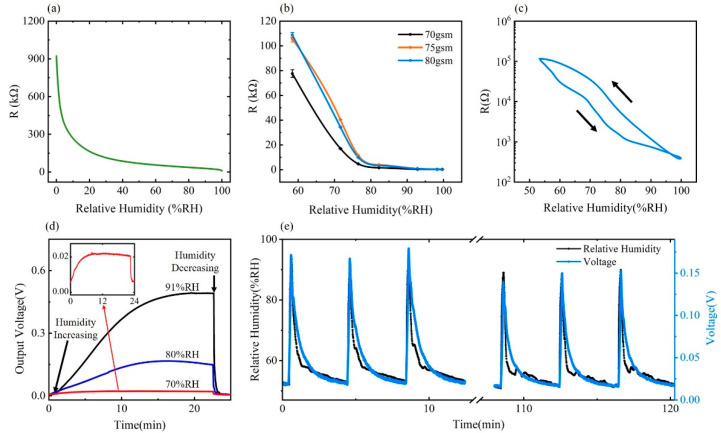
(**a**) Relationship between PHS resistance and relative humidity in paper humidity sensing model. (**b**) Resistance-RH% characteristic curve of PHSs under different design parameters. (**c**) Hysteretic curve of 80 gsm PHS. (**d**) Response time of 80 gsm PHS. (**e**) Repeatability of 80 gsm PHS.

**Figure 4 materials-15-06447-f004:**
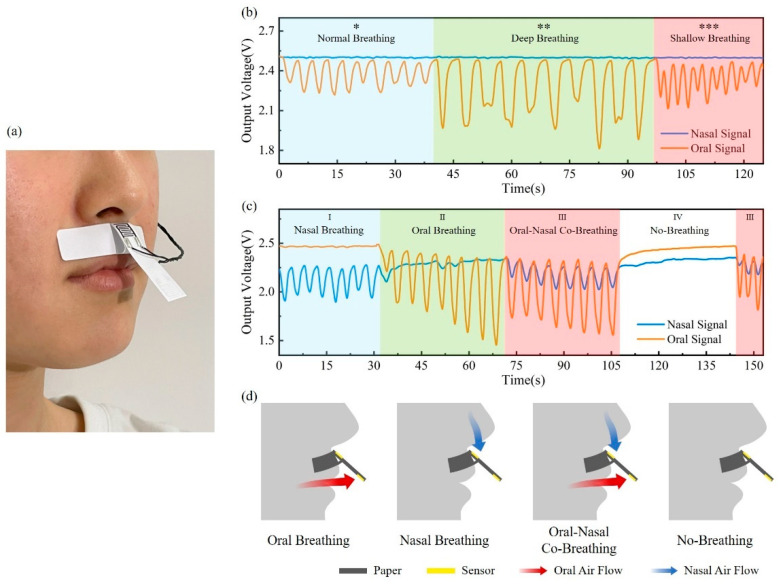
(**a**) Photo of a volunteer wearing a PHS patch. (**b**,**c**) Simulated breathing experiments for different breathing modes and various respiratory patterns (*: normal breathing; **: deep breathing; ***: shallow breathing). (**d**) Schematic diagram of oral and nasal airflows in four respiratory patterns.

**Figure 5 materials-15-06447-f005:**
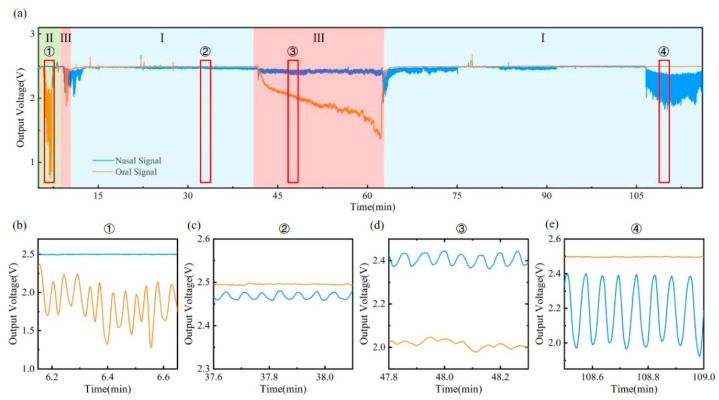
Sleep respiratory monitoring of healthy volunteer: (**a**) Breathing signals throughout the test (I: nasal breathing; II: oral breathing; III: oral–nasal co-breathing). (**b**–**e**) Detailed signals for different sleeping stages during the test, including oral breathing (stage ①), nasal breathing (stage ② and stage ④), and oral–nasal co-breathing (stage ③).

**Figure 6 materials-15-06447-f006:**
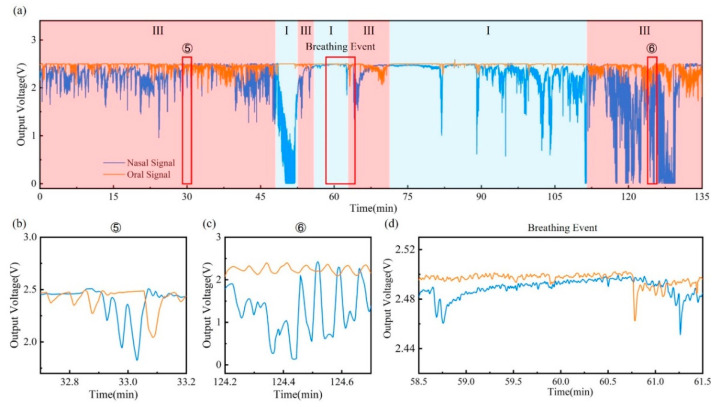
Sleep respiratory monitoring of apnea patient: (**a**) Breathing signals throughout the test (I: nasal breathing; III: oral–nasal co-breathing). Detailed signals for different sleeping stages during the test, including (**b**,**c**) oral–nasal co-breathing (stage ⑤ and stage ⑥) and (**d**) breathing event.

## Data Availability

Data are contained within the article.

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
