# Peer review of "Paper-Based Humidity Sensor for Respiratory Monitoring"

_materials, 2022, doi:10.3390/ma15186447_

Round 1

Reviewer 1 Report

See attached file.

Reviewer 2 Report

The authors report the construction of a paper-based device for monitoring nasal or oral breathing based on moisture in the exhaled breath. The devices are characterized in a humidity chamber, as well as two recordings from humans. Paper-based devices are argued to be cheaper and more convenient for home-based sleep analysis via tracking breathing. The performance of the devices is well documented, and mathematical models for the behavior are presented. In addition to the specific comments below, further discussion of the mechanisms underlying the process and model development would greatly strengthen the manuscript.

Line 56 mentions screen printing, Figure 1 mentions screen painting. Are these the same process?

Figure 1 – the text of the legend is very small. Can the wire and water be above/below each other and everything use a larger font?

Line 66-72 – what are the charge carriers in the reduced resistance sensors? Are charge carriers arriving from the absorbed moisture, or does moisture mobilize existing charge carriers?

It was not clear in the fabrication that the tape in Figure 1 is for attaching the sensor to the body instead of assembling the sensor. Some sentences in the first paragraph or including an image like from Figure 4 in Figure 1 could make this clearer.

Line 121 - remove ‘at the same time’ it is redundant.

Line 130-131 the sentence is unclear. Please re-word.

Section 2.1 please include the dimensions of the interdigitated electrodes so the resistance can be compared to the resistivity of water (the proposed charge carrier source).

Section 3.1 – ‘Fabric model’ may be mis-understood as a device on textile instead of paper. A name like, ‘Generation of the Model’, ‘Derivation of the Model’, or ‘Resistance Model’ would avoid confusion.

Line 170-175 – does paper thickness alter the humidity range of high sensitivity by trapping moisture deeper in the material between each breath? Or does thinner paper allow drying from the back of the sheet (full-thickness wetting)? Some comments are made regarding the performance of different thicknesses, but little is said about the mechanism.

Figure 3 (d) – the inset is too small to read the numbers.

Line 184 – these are very specific and unusual values. Do they represent specific known target humidities? Are the values explained by properties of exhaled breath such as temperature when exhaled, or humidity of exhaled breath under certain conditions? See also https://doi.org/10.1016/j.snb.2019.127371

Line 196 - please be specific ‘…within the targeted sensing range of X%-Y%.’ and provide a relevant citation for that being the range for breath sensing.

Line 201-207-217 – please consider re-writing this, possibly as one paragraph, including measurements relative to experimental details and the biologically relevant values. In Figure 3d, the voltage lags behind wetting but has a sudden drop during drying. However, in Figure 3e the voltage lags behind the humidity during drying and follows accurately during wetting. This effect should be described in a way that provides more analysis. Please explain the physics between the wetting vs. drying and why they involve different time constants. On a more practical level, it should be discussed if the response times are suitable for use with a system that changes humidity several times per minute (such as breathing). Since Figure 3 operates on the scale of minutes but breathing operates on less than one minute, it would be informative to know how the sensor responds when the drying period has not reached baseline between wetting. Are all the data points in Figure 3e necessary? Can a smaller picture still make the point? Or could stability be well described with a statistic in the text, such as testing if there is any significant difference between the first X peaks and the last X peaks. Could some of the Figure space be better used to show the %RH and V during short bursts such that the Figure has 6 plots in total?

Line 218-228 – Is there a difference between repeatability (cycle number) and durability (time in use)? Do sub-saturation durations change the matching rate by resulting in accumulating wetting?

Line 234-239 – please clarify if ‘simulated’ breathing is measured on a person who is consciously controlling their respiration, or if this is a device that simulates respiration, or if this is a mathematical model of the response to hypothetical breathing based on the humidity chamber data.

Line 265 ‘the paper-based sensor patch was stuck on the skin,…’

Line 274 – 14-20 is likely intended to be 14-20 per minute not per second (Hz). Please also provide a citation for this range being normal.

Figure 5 – in the caption please explain II, III, I, III, I.

Is there an explanation for the drifting signal in Figure 5a III ~45-60 minutes and the drifting signal in Figure 6a I ~47-52 minutes?

Line 317 – A large point is made about price reduction using this paper sensor. What is the expected percent of cost saving compared to the current market standard(s)? How does disposability affect the proposed cost reduction?

Reviewer 3 Report

1. Authors have to define OSA in the abstract.

2. What will be the impact on sensing, if the number of fingers for electrodes will increase?

 3. Authors have mentioned that the sensor is good for humidity sensing. In the presence of any gas how it will differentiate?

4. How to improve the response time of the sensor because in the current situation its quite high. 
